# Outer retinal features in OCT predict visual recovery after primary macula-involving retinal detachment repair

Christof Hänsli[1]*, Suijana Lavan[2], Isabel B. Pfister[1], Christin Schild[1], Justus G. Garweg[1,2,3]

1 Berner Augenklinik am Lindenhofspital, Bern, Switzerland, 2 Medical Faculty, University of Bern, Bern, Switzerland, 3 Department of Ophthalmology, Inselspital, University of Bern, Bern, Switzerland

* christof.haensli@gmail.com

**Data Availability Statement:** All relevant data are within the manuscript and its Supporting information files.

## Abstract

### Purpose

To find predictive markers for the visual potential in optical coherence tomography (OCT) one month after surgical repair of macula-involving rhegmatogenous retinal detachment (miRD) with and without internal limiting membrane (ILM) peeling.

### Methods

This retrospective single-center, single-surgeon cohort study included 74 patients who underwent pars plana vitrectomy (PPV) for primary miRD between January 2013 and August 2020 with follow-up examinations for at least 6 months. Patients developing recurrent detachments, media opacities, or with an axial length over 27 mm were excluded from the analysis. LogMAR visual (VA) and LogRAD reading acuity (RA) ± standard deviation (SD), and OCT measurements 6 months after surgery were compared to OCT and VA measurements one month after surgery using multiple linear regression analysis for predictions.

### Results

VA increased from 0.34 ± 0.25 at one month to 0.22 ± 0.21 after 6 months [$p < 0.001$; effect size = -0.662, 95% confidence interval (CI): -(0.99–0.33)]. The continuity of the external limiting membrane (ELM) and ellipsoid zone (EZ) increased between 1 and 6 months. Subfoveal ELM integrity after one month predicted VA [adjusted $R^2$ of 8.0%, $F_{(2, 71)} = 4.17$, $p = 0.018$] and RA [adjusted $R^2$ of 29%, $F_{(2, 27)} = 6.81$, $p = 0.002$] after 6 months. EZ integrity had a less pronounced predictive effect on VA and RA. ELM integrity after 1 month correlated with better reading acuity after 6 months ($p = 0.016$).

### Conclusion

VA and morphological OCT parameters improve between 1 and 6 months after surgery for miRD. The grade of ELM is a better predictor for RA than for VA, explaining more variance.

**Funding:** The author(s) received no specific funding for this work.

**Competing interests:** The authors have declared that no competing interests exist.

## Introduction

Macula-involving rhegmatogenous retinal detachment (miRD) is accompanied by a significant risk of permanent vision loss despite successful re-attachment surgery. Preoperative vision, duration of vision loss, and number of retinal breaks have been reported as preoperative factors associated with the postoperative visual potential, but published data are contradictory for some of these factors [1, 2]. Preoperative features of optical coherence tomography (OCT) such as photoreceptor length, height of foveal detachment, inner retinal undulation and inner retinal separations have been discussed as biomarkers of visual recovery [3, 4]. In cases of foveal detachment, visual recovery seems better if the macula is only partially detached, compared to a complete macular detachment [5]. Moreover, a preserved foveal depression despite foveal detachment seems predictive of a better functional outcome [6]. The optimal timing and technique of surgery are subject to discussion, and peeling of the internal limiting membrane (ILM) has been discussed as a potential factor for improved postoperative visual recovery, with evidence being still unclear [7–10]. A meta-analysis of available retrospective studies did not unequivocally support this finding [11]. A structural correlation of the fovea and visual recovery after vitrectomy has not been made to date.

The initial aim of this retrospective study was to compare ILM peeling to no ILM peeling, together with predictive imaging parameters in early postoperative OCT after 1 month. Due to high subgroup imbalance with few and biased cases without ILM peeling, we focused on early postoperative predictive biomarkers. The aim of this retrospective study was therefore to test a possible correlation of the aforementioned pre- and early postoperative features with the visual outcome after 6 months, and to compare both the postoperative OCT features and the visual recovery after successful re-attachment surgery.

## Patients and methods

This retrospective study draws on consecutive data from patients undergoing surgical retinal detachment repair who were treated within 48 hours of referral at our institution by a single surgeon between January 2013 and August 2020. The study followed the tenets of the Declaration of Helsinki and was approved by the local ethical committee (BASEC-ID 2020–02920), with informed consent for analysis and publication of data from all patients. Inclusion criteria were primary rhegmatogenous retinal detachment with macular involvement (miRD), treated with pars plana vitrectomy, endodrainage, endolaser, and gas tamponade. Exclusion criteria were a postoperative follow-up of under 6 months, use of another tamponade, i.e. silicone oil, recurrent retinal detachment and/or any ocular surgery within the follow-up period of 6 months, cataract with significant visual impairment (if not treated during re-attachment surgery), age under 18 years at time of surgery, an axial length longer than 27 mm, active systemic and ocular inflammatory and vascular diseases requiring treatment (i.e. active rheumatological and metabolic diseases including diabetes, uveitis and retinal vascular occlusion), and finally, denial of informed consent. In patients with retinal detachment in both eyes, only the first-affected eye was included.

Data were collected retrospectively from the electronic patient records at baseline before surgery (BL), and 1 month, 6 months, and where available 12 months after surgery. This included best corrected visual acuity (BCVA) using Snellen decimal charts, reading acuity (RA), axial length, duration of visual impairment, and optical coherence tomography (OCT; Heidelberg Spectralis spectral-domain OCT with 880 nm wavelength, with axial resolution of 3.9 μm and lateral resolution of 5.7 μm, Heidelberg Engineering, Heidelberg, Germany). Assessment of OCT included preoperative presence of outer retinal separation and undulation, foveal height of retinal detachment as the height of the subfoveal space, and

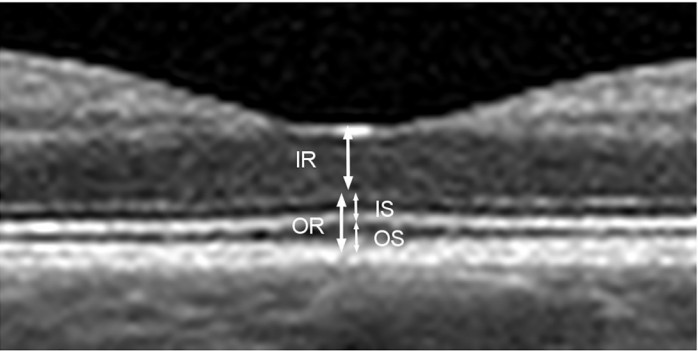

**Fig 1. Retinal layers.** Central foveal cross-section of optical coherence tomography (OCT) of a normal eye illustrating the measurements: Inner retina (IR), from the internal limiting membrane to the inner margin of the external limiting membrane (ELM). Inner segment (IS) of the photoreceptor layer, from the inner margin of the ELM to the outer margin of the ellipsoid zone (EZ). Outer segment (OS) of the photoreceptor layer, including the outer photoreceptor segment and the interdigitation zone. The outer retina (OR) is composed of IS and OS together.

photoreceptor outer segment thickness. One, 6, and 12 months after surgery, BCVA as well as OCT data were recorded. OCT assessments included manual grading of inner and outer retinal thickness, and inner and outer photoreceptor segment thickness, as well as outer photoreceptor segment thickness of the other eye.

Measurements of the OCT images were performed on the central horizontal B-scan at the foveal pit. Measurements of inner (IS) and outer (OS) photoreceptor segments, as well as inner (IR) and outer (OR) retinal height were assessed in the central fovea (Fig 1). The continuity of the external limiting membrane (ELM) and the ellipsoid zone (EZ) under the foveal pit were graded from one to four: 1 normal and continuous, 2 altered but continuous, 3 interrupted but recognizable, and 4 not recognizable (Fig 2). At baseline, wherever OCT was available, foveal detachment height was measured from the retinal pigment epithelium to the outer border of the retina, and presence or absence of retinal separation and undulation was assessed.

Data collection and OCT measurements were performed in a non-blinded manner by a single author (SL) and verified by another author (CH). Statistical analysis was performed using R (R Core Team, 2021. R: A language and environment for statistical computing. R Foundation for Statistical Computing, Vienna, Austria. http://www.R-project.org/). For statistical purposes, BCVA was converted to the logarithm of the minimum angle of resolution (LogMAR), RA was converted to the logarithm of the reading acuity determination (LogRAD). All values are reported as mean with standard deviation (SD) as well as median and interquartile range (IQR). Changes over time are analyzed using one-way repeated measurements ANOVA for normally distributed data or the Friedman test for related data in cases where the data were not normally distributed. Significance was set at $p < 0.05$, and reported with effect sizes as well as lower and upper 95% confidence intervals (CI). The relationship between measurements was evaluated using multiple regression. Independence of observations was tested by applying the Durbin-Watson statistic.

# Results

## Functional outcomes

Of a total of 406 surgeries for retinal detachment in the index period, 317 showed macular involvement, 228 were treated with PPV and sulfur hexafluoride (SF6) gas, and one each with

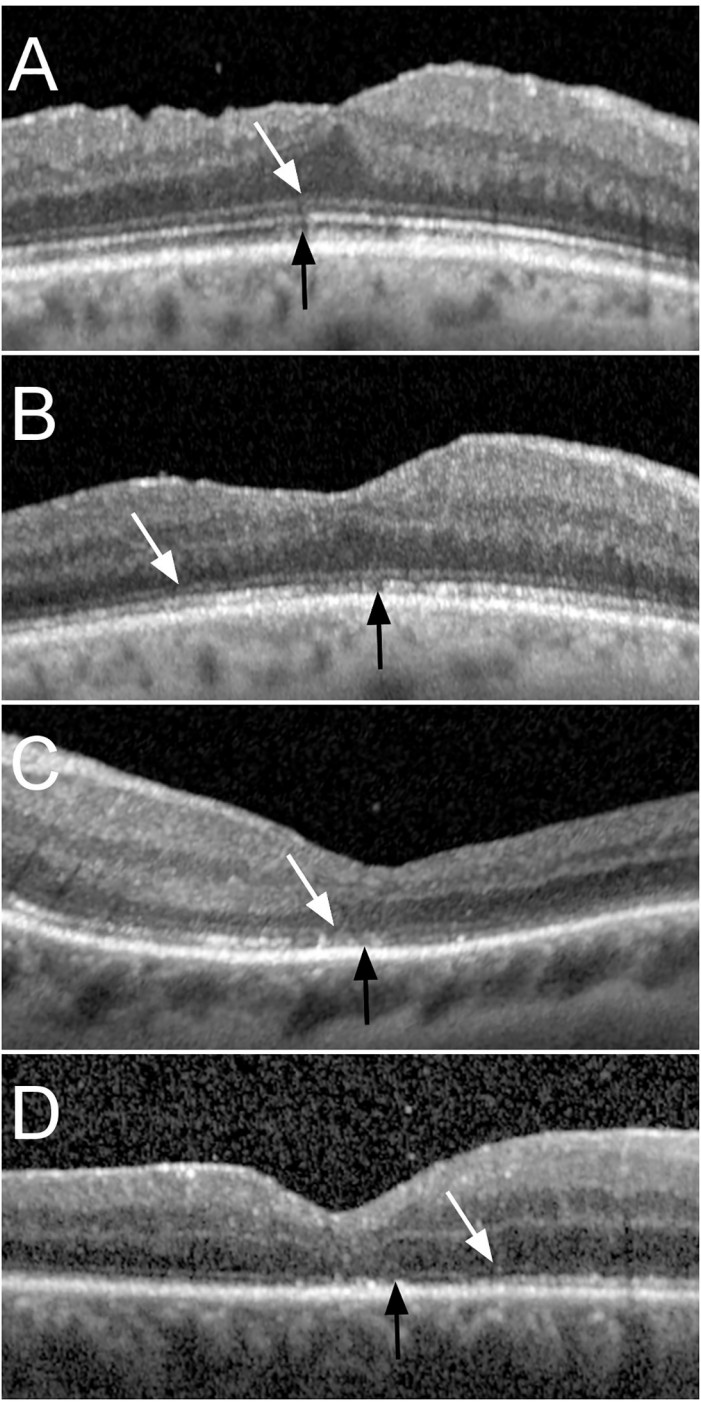

**Fig 2. ELM and EZ grading.** Optical coherence tomography (OCT) images of different eyes representative of the different severity grades of external limiting membrane (ELM) and ellipsoid zone (EZ) impairment. For both structures, grade 1 is defined as a normal and continuous structure, grade 2 as altered but continuous, grade 3 as interrupted and grade 4 as absent. A: ELM and EZ grade 1: Fovea 6 months after re-attachment surgery, where ELM and EZ are continuous and only show regular alterations comparable to normal eyes, with a visual acuity of 0.2 LogMAR. B: ELM and EZ grade 2: ELM and EZ are continuous with subfoveal alterations—very minor in the case of ELM—1 month after surgery, with a visual acuity of 0.4 LogMAR (same patient as in A). C: ELM grade 3 and EZ grade 3, the ELM shows small interruptions and the EZ alterations in the foveal area, 1 month after retinal detachment surgery, with a visual acuity of 0.3 LogMAR. D. ELM grade 3 and EZ grade 4: the ELM (arrow) is interrupted and the EZ is terminated outside the foveal area (arrowhead), 1 month after retinal detachment surgery, with a visual acuity of 0.4 LogMAR.

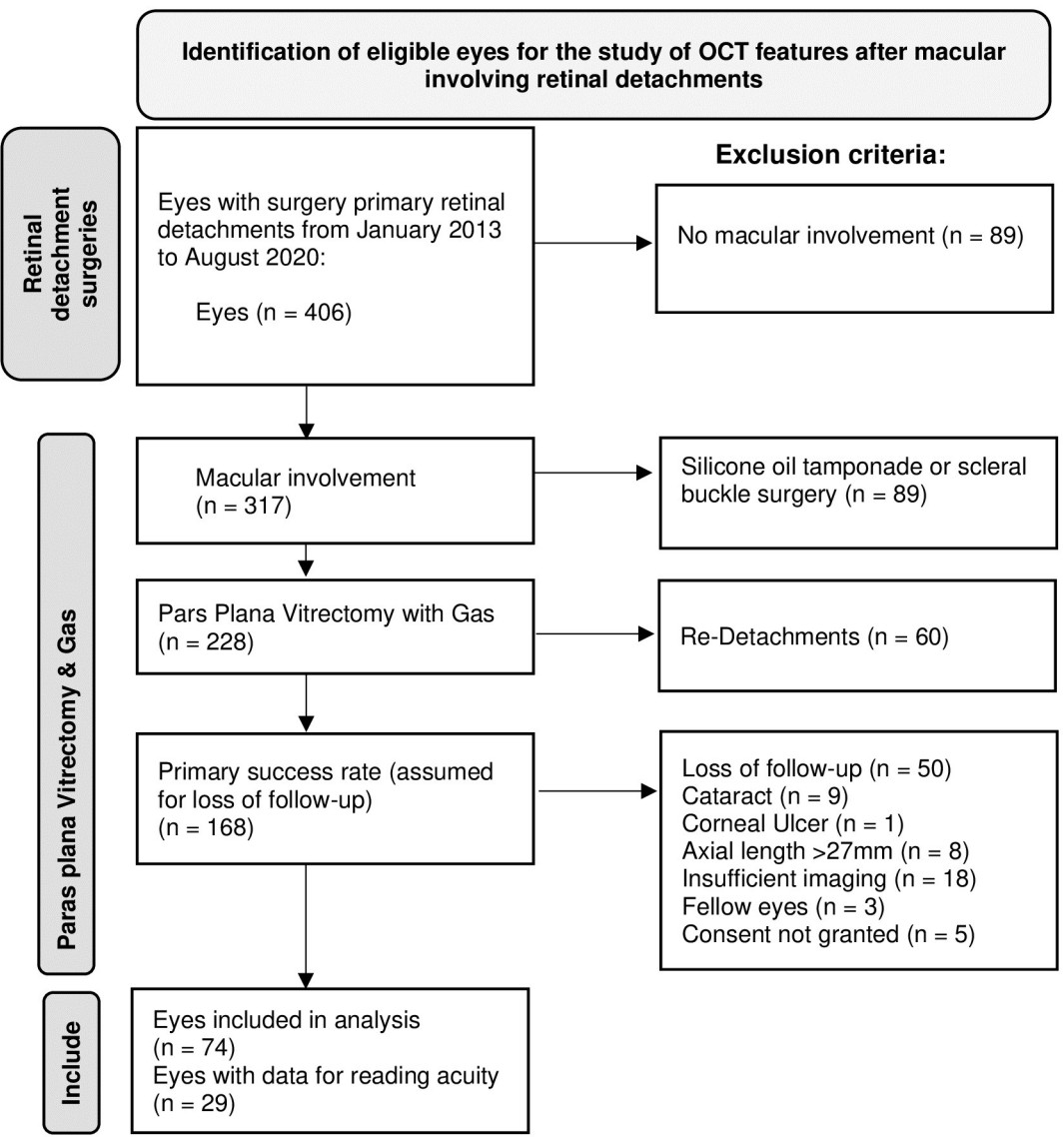

**Fig 3. Inclusion and exclusion criteria.** Flow chart illustrating the inclusion and exclusion criteria for analysis. Starting from a total of 406 eyes with primary rhegmatogenous retinal detachment from January 2013 to August 2020, 317 eyes showed macular involvement, and 228 of these underwent pars plana vitrectomy with SF6 or C3F8 gas endotamponade, with 168 primary successes after 6 months.

perfluoropropane (C3F8) and air tamponade. Sixty patients (26%) had to be excluded due to re-detachment within 6 months. Of the remaining 168 eyes, only 74 were included in the analysis, mainly due to loss of follow-up (n = 50), with the complete list of reasons shown in Fig 3. Of the 74 included eyes, 68 (92%) underwent ILM peeling, and six (8%) did not. Twenty-two patients (30%) were female, and the mean age of the patients was 65.4 ± 10.1 years. Complete foveal detachment was present in 64 eyes (86.5%) from the included sample. The number of patients without complete foveal detachment (13.5%) was too small to allow a statistical comparison between patients with and without this condition regarding visual acuity and its development over time. Furthermore, these subgroups show different baseline characteristics, introducing relevant bias into the comparison.

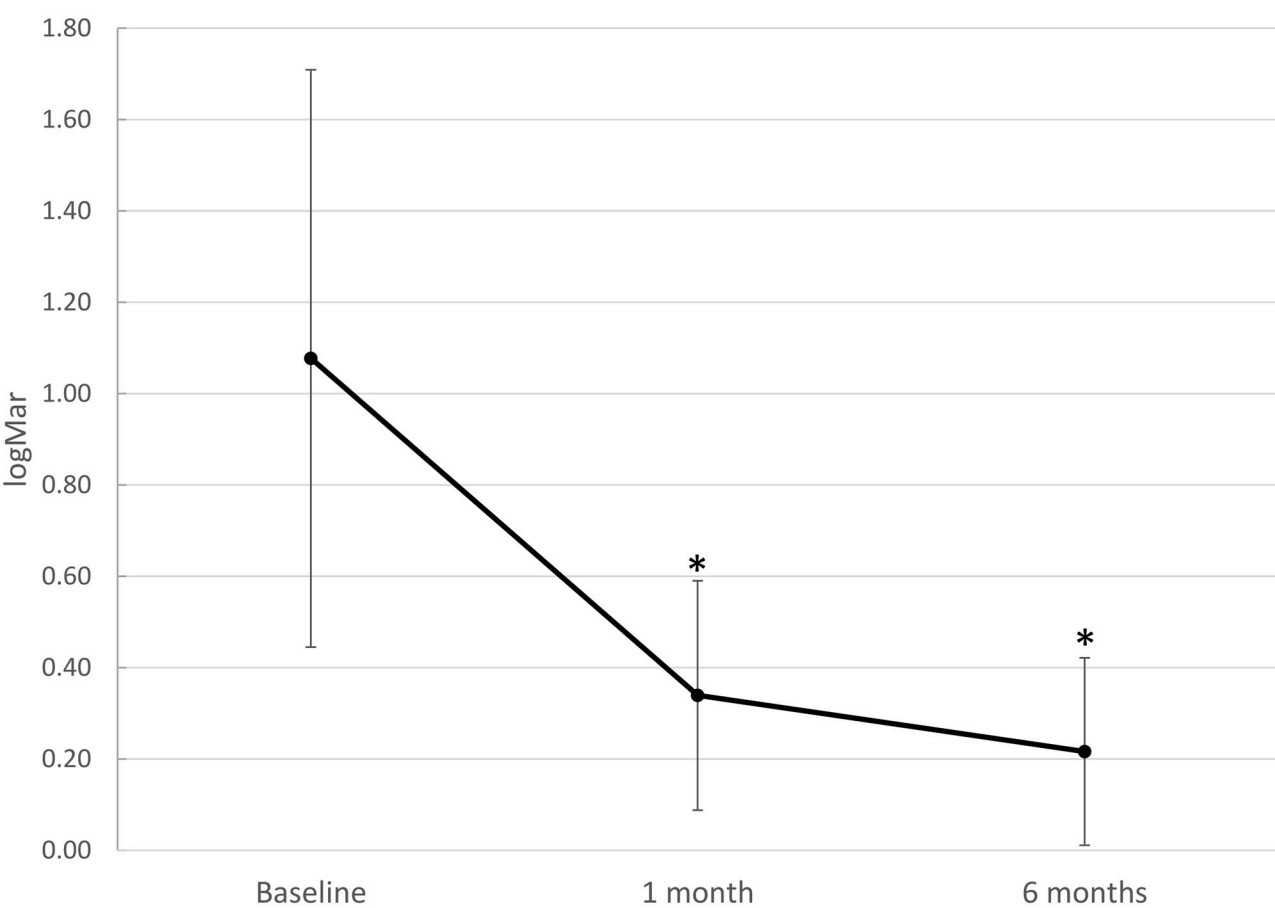

**Fig 4. Visual acuity recovery.** Visual acuity (VA) indicating progressive visual recovery from baseline before vitrectomy for re-attachment of macula-involving primary rhegmatogenous retinal detachment with vitrectomy and gas. Baseline VA (LogMAR) improved after 1 and 2 months. Whiskers indicate standard deviations and asterisks (*) indicate significant differences compared to baseline (p < 0.001; n = 74 for all three time points).

**Visual acuity.**   At baseline, a mean BCVA of 1.08 LogMAR (SD ±0.63; median 1.2, IQR 0.6 to 1.5 LogMAR) was recorded. Visual acuity (VA) gradually increased to a mean of 0.34 LogMAR (SD ±0.25, median 0.3, IQR 0.2 to 0.4) after 1 month [effect size: $d_{\text{Repeated Measures}}$ = 0.995, 95% CI: -(1.34–0.65)]. VA further increased to 0.22 LogMAR [SD ±0.21, median 0.2, IQR 0.1 to 0.3, effect size: $d_{\text{Repeated Measures}}$ = 1.104, 95% CI: -(1.45–0.76)] after 6 months for the entire study sample. For both time points, the increase in visual acuity was significant compared to baseline (Friedman test for correlated samples $p < 0.001$; Fig 4), as was that from 1 to 6 months [$p < 0.001$; effect size: $d_{\text{Repeated Measures}}$ = 0.662, 95% CI -(0.99–0.33)].

**Reading acuity.**   Reading acuity (RA) measurements were available for only 11 patients at baseline, and 29 eyes after 1, and 30 eyes after 6 months. Since the amount of available data at baseline was small and biased towards better results, we refrain from making any statements about this time point. Reading acuity increased from 1 month (mean 0.48 LogRAD, SD ±0.21, median 0.5, IQR 0.4 to 0.6), to 6 months [0.37 LogRAD, SD ±0.28, median 0.35, IQR 0.1 to 0.5; normally distributed data: t-test for paired samples (instead of ANOVA for paired samples, since we have only two time points): $p = 0.026$; effect size: $d_{\text{Repeated Measures}}$ = 0.645, 95% CI: -(1.17–0.12) for the pooled sample] (Fig 5).

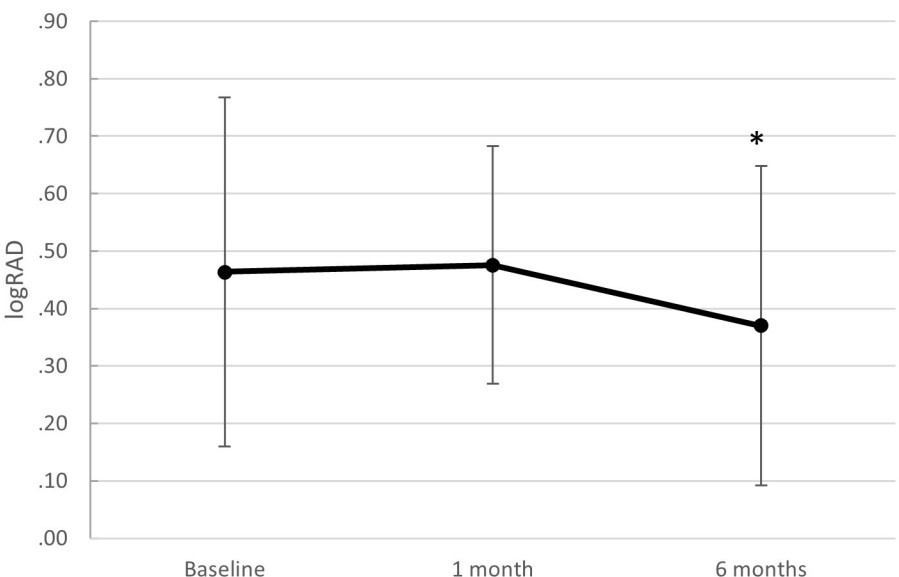

**Fig 5. Reading acuity recovery.** Reading acuity (RA) from baseline before surgical repair of macula-involving primary rhegmatogenous retinal detachment with vitrectomy and gas. The small and variable sample sizes with possible bias may explain the lack of statistical significance. Whiskers indicate standard deviations and the asterisk (*) indicates a significant difference compared to 1 month (p = 0.026); n = 11 at baseline, n = 29 after 1 month, and n = 30 after 6 months.

## Morphological outcomes

Morphological parameters in the central OCT B-Scan were measured for inner retina, outer retina, and the inner and outer segments of the photoreceptors (together forming the outer retina). Recovery of the foveal structure can be observed visually (Fig 6).

**Inner retina (IR) and outer retina (OR).** Mean IR thickness was 219.7 μm (SD ±79.8, median 216, IQR 156 to 164.5) after 1 month, and decreased to 206.2 μm (SD ±57.6, median 209, IQR 254.5 to 239.5) after 6 months (effect size: $d_{\text{Repeated Measures}}$ = 0.244, 95% CI: -0.57–0.081; Friedman test for correlated samples $p$ = 0.014). Mean OR thickness was 75.3 μm (SD ±19.5, median 75, IQR 61.8 to 83.5) after 1 month, and increased to 83.9 μm (SD ±11.8, median 82, IQR 78 to 89) after 6 months (effect size: $d_{\text{Repeated Measures}}$ = 0.409, 95% CI: 0.081–0.737; Friedman test for correlated samples $p$ < 0.001, Fig 7).

**Inner segment (IS) and outer segment (OS) height of the photoreceptor layers.** Mean IS increased from 38.3 ± 9.6 μm (median 38.0, IQR 32 to 44) after 1 month to 43.0 ± 7.1 μm (median 43, IQR 39 to 48) after 6 months (effect size: $d_{\text{Repeated Measures}}$ = 0.43, 95% CI: 0.10–0.76; Friedman test for correlated samples $p$ = 0.004). At 1 month OS was 37.2 ± 14.5 (median 34.5, IQR 28 to 41) and showed a non-significant increase from 1 to 6 months (41.0 ± 8.9, median 39.0, IQR 35 to 45.5; effect size: $d_{\text{Repeated Measures}}$ = 0.24, 95% CI: -0.20–0.68, Friedman test for correlated samples $p$ = 0.13; Fig 8).

**Recovery of the External Limiting Membrane (ELM).** After 1 month, ELM was grade 1 in 14 (18.9%), grade 2 in 36 (48.6%), grade 3 in 23 (31.1%), and grade 4 in one patient (1.4%). ELM improved to grade 1 in 47 (63.5%), grade 2 in 23 (31.1%), and grade 3 in three eyes (4.1%) 6 months after surgery, while no patients were grade 4. One patient underwent no OCT imaging after 6 months (Fig 9).

**Ellipsoid Zone (EZ).** One month after surgery, EZ was grade 1 in three (4.1%), grade 2 in 43 (58.1%), grade 3 in 24 (32.4%) and grade 4 in four (5.4%) of eyes, and changed to grade 1 in

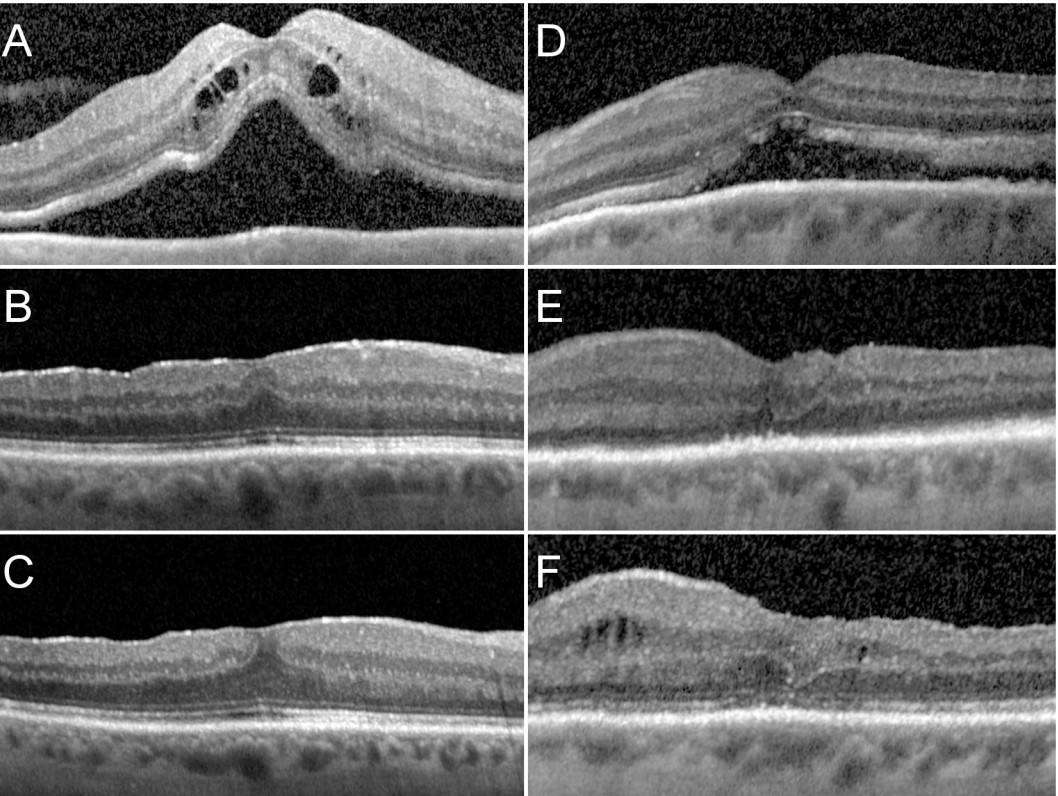

**Fig 6. Exemplary evolution in two instances.** Left column A–C: A patient with foveal detachment reporting visual loss for 14 days and a preoperative visual acuity (VA) of 0.7 logMAR and a reading acuity (RA) of 0.3 logRAD before surgery (A), with good visual and morphologic recovery (ELM 1, EZ 2, VA -0.1, RA 0.0) after 1 month (B), and further morphological improvement after 6 months (ELM 1, EZ 1, VA -0.1, RA 0.0) (C). Right column D–F: A patient with foveal detachment reporting visual loss for 30 days (D), with limited early functional recovery (VA 0.9 logMAR, RA 0.7 logRAD) and morphological improvement (ELM 3, EZ3) (E), and slight but still incomplete recovery after 6 months (VA 0.7 logMAR, RA 0.7 logRAD) and persisting structural alterations (ELM 2, EZ 2) (F).

31 (41.9%), grade 2 in 36 (48.6%), and grade 3 in six (8.1%) eyes after 6 months. Grade 4 was not seen at this time point.

## Multiple linear regressions for predictive factors for visual recovery

Multiple linear regression models were used to VA (n = 74) and RA (n = 29) after six months as dependent variables. As independent variables and predictors we used only morphological parameters for model 1 and morphological and functional parameters for model 2 (Table 1).

Model 1 included ELM and EZ grade after 1 month. Model 2 included VA after one months, and for prediction of RA, also RA after 1 month.

In the models analyzing effects on VA after 6 months (n = 74), we see a small yet significant effect of ELM grade after 1 month in predicting final VA [adjusted $R^2$ of 8.0%, F(2, 271 = 4.17, p = 0.018), if only morphological parameters are used. With the inclusion of VA after 1 month, the model's total adjusted $R^2$ achieved a medium effect of 52%, achieving statistical significance (p = 0.017). In the models analyzing effects on RA after 6 months (n = 29), we see a significant medium-size effect of ELM grade after 1 month in predicting final RA [adjusted $R^2$ of 29%, F(2, 27) = 6.81, p = 0.002], if only morphological parameters were used. With the inclusion of

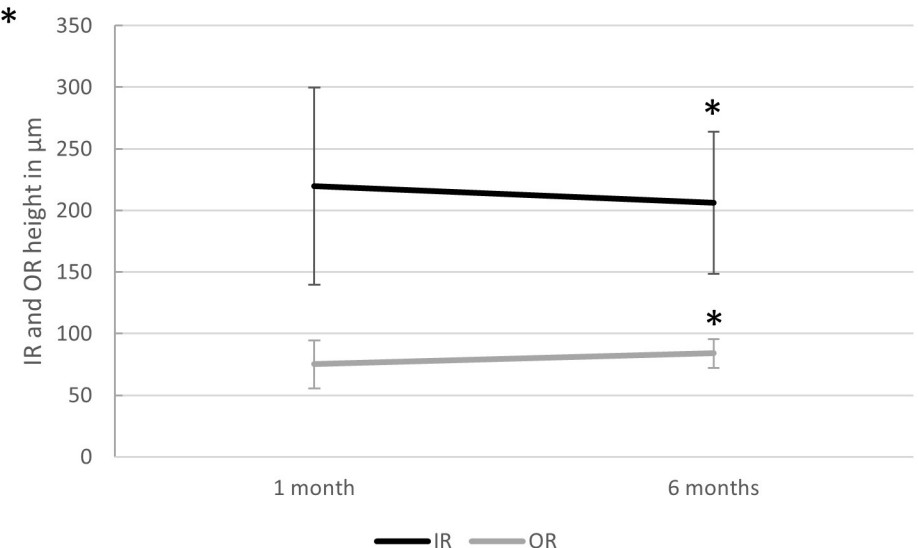

**Fig 7. Inner and outer retinal height changes.** Change in subfoveal inner (IR) and outer (OR) retina height 1 and 6 months after vitrectomy with gas endotamponade for macula-involving primary retinal detachment. A slight but significant decrease in IR and increase in OR were seen. Mean IR thickness decreased from 219.7 ± 79.8 μm to 206.2 ± 57.6 μm, and OR increased from 75.3 ± 19.5 μm to 83.9 ± 11.8 μm (Friedman test for correlated samples p = 0.014 for IR, and p = 0.0005 for OR, n = 74 at 1 month and n = 73 at 6 months). Whiskers indicate standard deviations, and asterisks (*) indicate significant differences compared to 1 month (p < 0.01).

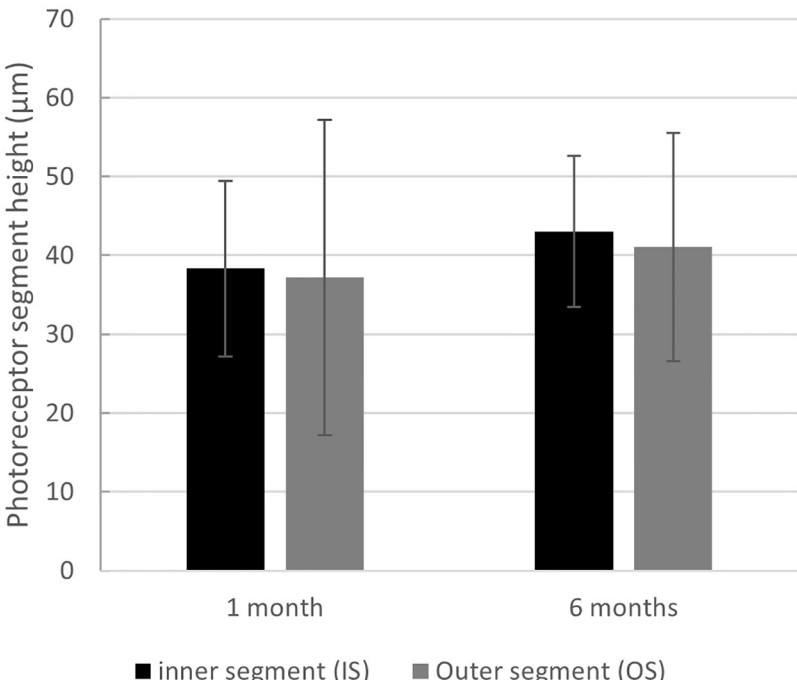

**Fig 8. Inner and outer photoreceptor height changes.** Change in subfoveal inner (IS) and outer (OS) photoreceptor segment height 1 and 6 months after vitrectomy with SF6 gas endotamponade for macula-involving primary retinal detachment. A slight increase in both IS and OS heights and no obvious changes in the IS/OS ratio may indicate a lack of clinical relevance (p > 0.05 for both). Whiskers indicate standard deviations.

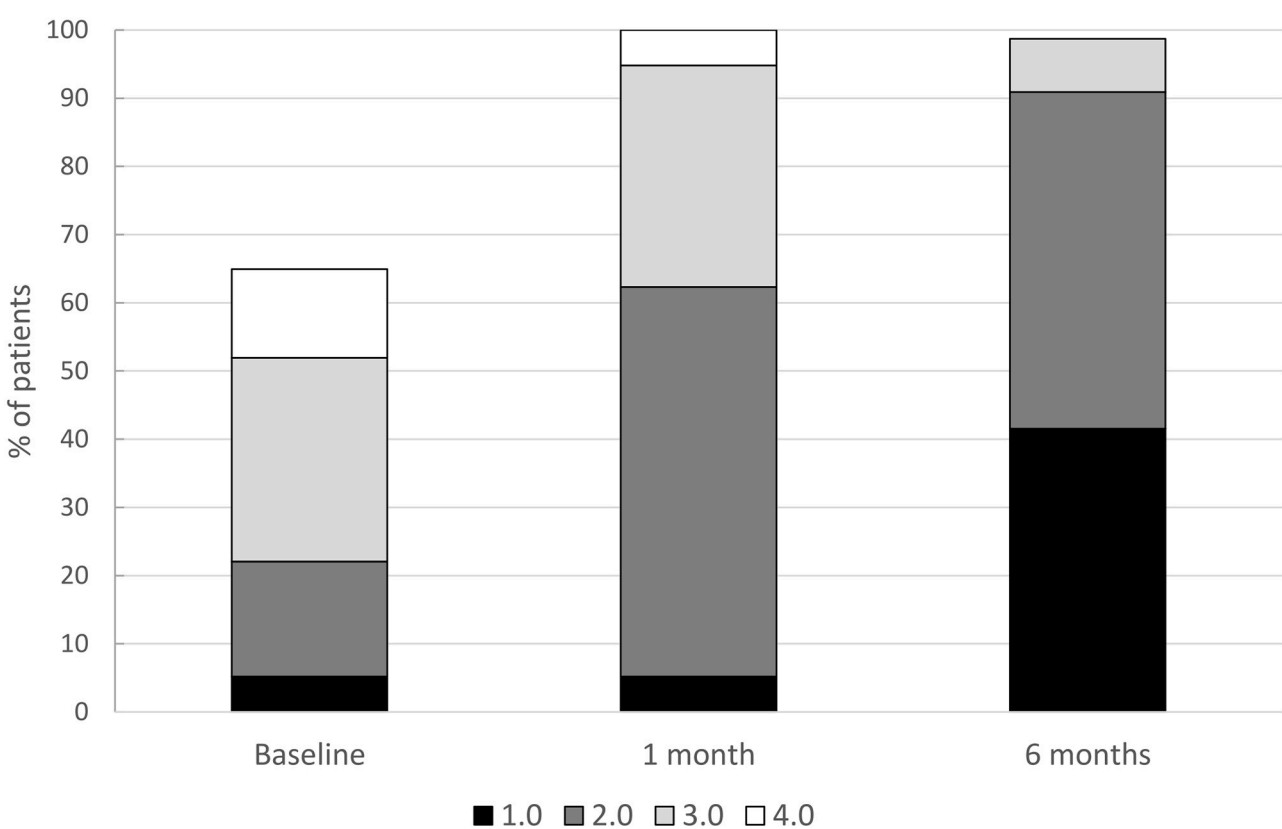

**Fig 9. ELM grade change.** Change in ELM severity grade after surgical repair for macula-involving rhegmatogenous retinal detachment (miRD). ELM was qualitatively graded in the subfoveal OCT into grade 1: normal and continuous; grade 2: altered but continuous; grade 3: interrupted but recognizable; and grade 4: not recognizable. The graph shows the qualitative recovery of the ELM in OCT imaging after vitrectomy for miRD.

VA and RA after 1 month, the model's total adjusted $R^2$ was 44%, without individual statistical significance for the included values. In other words, 8% of the variance in VA is predicted by ELM and EZ grade 1 month postoperatively ($p = 0.019$), and to 52% if VA was also included ($p < 0.0001$). Reading acuity after 6 months is explained to 29% by grade of ELM and EZ at 1 month ($p = 0.004$), and to 44% if VA and RA are also included.

When corrected for age, gender, ILM peeling, or presence of foveal detachment, the overall models including one-month grades of ELM with or without VA and RA remained statistically significant for both final VA and RA (S3 Table).

Separate analyses were performed with a simplified ELM grading into continuous (grades 1 and 2) or interrupted ELM (grades 3 and 4). The analyses revealed no predictive value of ELM integrity (grades 1–2 versus grades 3–4 taken together) on better or worse VA after 6 months ($p = 0.51$, n = 74). ELM integrity, however, correlated with RA after 6 months ($p = 0.016$, n = 29). Thus, ELM grades 1 or 2 demonstrated a better RA after 6 months compared to grades 3 or 4.

Furthermore, multiple regression models were developed using one-month measurements of photoreceptor inner (IS) and outer (OS) segment height with VA and RA at 6 months as dependent variables (S3 Table). Photoreceptor IS and OS had an adjusted $R^2$ of 8% on final VA ($p = 0.024$), and of 15% on final RA (p = 0.04). A composite model of presence of foveal detachment, detachment height, retinal separation and undulations before surgery did not yield significance on final VA or RA.

**Table 1. Results of the multiple linear regression analyses.**

| | β | t | p | F | df | p | adj. $R^2$ |
|---|---|---|---|---|---|---|---|
| Dependent variable: Visual acuity at 6 months | | | | | | | |
| Model 1 | | | | 4.17 | 2 | 0.019* | 0.08 |
| ELM 1 month | 0.10 | 2.43 | 0.018* | | | | |
| EZ 1 month | -0.02 | -0.47 | 0.64 | | | | |
| Model 2 | | | | 27.84 | 3 | 5.83e-12** | 0.52 |
| ELM 1 month | -0.01 | -0.37 | 0.71 | | | | |
| EZ 1 month | 0.006 | 0.18 | 0.85 | | | | |
| VA 1 month | 0.62 | 8.21 | 7.56e-12*** | | | | |
| Dependent variable: Reading acuity at 6 months | | | | | | | |
| Model 3 | | | | 6.81 | 2 | 0.004 | 0.29 |
| ELM 1 month | 0.26 | 3.34 | 0.002** | | | | |
| EZ 1 month | -0.06 | -0.61 | 0.54 | | | | |
| Model 4 | | | | 5.69 | 4 | 0.003 | 0.44 |
| ELM 1 month | 0.08 | 1.09 | 0.29 | | | | |
| EZ 1 month | -0.002 | -0.02 | 0.99 | | | | |
| VA 1 month | 0.20 | 0.74 | 0.47 | | | | |
| RA 1 month | 0.48 | 1.68 | 0.11 | | | | |

ELM: external limiting membrane; VA: visual acuity; RA: reading acuity.

* $p < 0.05$,

** $p < 0.01$,

*** $p < 0.001$.

Multiple linear regression analyses of visual acuity (VA, n = 74) and reading acuity (RA, n = 29) as dependent variables, 6 months after surgical repair of macula-involving primary rhegmatogenous retinal detachment with pars plana vitrectomy and SF6 gas endotamponade. Model 1 included as morphological parameters 1 month postoperatively the qualitative grade of the external limiting membrane (ELM) and ellipsoid zone (EZ). Model 2 additionally included RA (n = 29) and VA (n = 74) assessments 1 month after surgery. Of final VA after 6 months, 8% of the variance is predicted by ELM and EZ grade 1 month postoperatively ($p = 0.019$), and to 59% if VA and RA are also included ($p < 0.0001$). Reading acuity after 6 months is explained to 29% by one-month grades of ELM and EZ ($p = 0.004$; Model 3), and to 44% if VA and RA are also included (Model 4). β = beta coefficient, the degree of change in the outcome variable for every 1-unit change in in the predictor variable; t = t statistic; p = p-value; F = F-statistic; df = degrees of freedom; adj. $R^2$ = adjusted $R^2$, the percentage of variation explained by the model.

## Discussion

Our results demonstrate that ELM is a better predictor of RA than VA, explaining more variance. ELM grade was able to explain 8% of the variance in VA compared to 29% in RA after 6 months. Also, a continuous ELM after 1 month is correlated with better RA after 6 months, compared with an interrupted ELM. While VA after 1 month could predict VA after 6 months, RA failed to predict either VA or RA after 6 months.

Our results demonstrate a surprisingly good overall visual outcome, with a mean BCVA of 0.22 LogMAR (20/33 Snellen equivalent) and 75% of patients below 0.3 LogMAR (better than 20/40 Snellen equivalent) 6 months after retinal detachment with macular involvement. We found a primary anatomical success rate of 74.7% in the group treated with SF6 (one each with C3F8 and air) gas tamponade, and of 81.1% in the group overall. In the underlying dataset, we used gas tamponade in 71.9% of 317 patients with macula-involving retinal detachment due to our policy of using gas endotamponade over silicone oil whenever possible, to avoid silicone-oil associated visual deteriorations. All surgical endpoints are within the range of published data, on the better side for visual function and on the poorer side for primary surgical success after 6 months, compared to studies combining vitrectomy and buckling surgery [12–16].

A gradual increase of visual and reading acuity from 1 to 6 months after surgery was linked to a qualitative morphological improvement of ELM and EZ in OCT B-Scan images, as well as a decrease in inner retinal thickness and an increase in outer retinal thickness. These findings are in accordance with previously published results, describing similar visual recovery and morphological retinal layer changes for up to 12 months with the major improvements occurring in the first 6 months [17–19]. On the other hand, we did not find distinct changes in outer photoreceptor segment length or inner/outer segment ratio.

The ELM grade, as the strongest and most consistent morphological predictor of final visual and reading acuity, can be quickly assessed from a single central B-Scan OCT, without the need for measurements of photoreceptor height or the memorization of normal ranges. The integrity of the ELM (grade 1 or 2 after 1 month) may thus be used as a rapidly assessable OCT biomarker for favorable reading acuity after 6 months, compared to cases with interrupted ELM (grades 3 or 4 1 month postoperatively).

## Limitations

The main limitations of our study include its retrospective nature and the limited sample size; both are within the range of the existing literature, but may biasing to some extent the quantitative outcomes reported, if not the qualitative ones. Our OCT measurements are limited to the central transfoveal B-Scan, and performed manually in a non-blinded manner. Grading of ELM and EZ seems simple and straightforward, but may be subject to a degree of inter- and intra-rater variability. We addressed this using a seniority-based voting system for all ambiguous OCT parameters. Transverse imaging of ELM and EZ for quantifiable analysis, as has been used in age related macular degeneration [20], was not feasible in our retrospective data set, as the scan distance was too wide to reconstruct transverse imaging.

The majority of our patients (87%) received ILM peeling, which makes it impossible to compare the ILM peeling group with the non-peeling group. This bias is explained by the personal experience of our surgeon (JGG), with improved visual outcomes after ILM peeling [7, 9]. A minority (14%) of patients showed an attached foveal center, with better functional prognosis. Functional parameters of the respective subgroups are presented in the supplementary material (S1 and S2 Tables). The comparison showed systematic bias towards less ILM peeling in the attached fovea group, and vice versa with better baseline VA. Thus, a subgroup analysis was not appropriate for these data, which was the primary intention of the study. The sample without peeling was too small to allow us to make meaningful statements about the differences between the groups with and without peeling. Henceforth, we describe the pooled data of both groups. A structural correlation between the fovea and visual recovery after vitrectomy with or without ILM peeling has not been made to date. The decision to analyze the pooled data is based on current evidence that neither a previous meta-analysis [11] nor a more recent retrospective study [8] could validate differences in functional or morphological outcomes between eyes with or without ILM peeling. Based on our primary study intention, we also included patients without peeling and pooled the results, though this may bias towards better final visual function and lesser functional improvement due to better visual function at baseline. Whether our findings can be applied on patients without ILM peeling remains unclear. Structural changes related to ILM peeling have been shown after retinal detachment surgery [21]. Our study cannot answer whether the observed structural changes in our cohort are due to the retinal detachment or the ILM peeling, which may longitudinally alter structure and thickness. Further limitations are the high drop-out rate of two-thirds of the possible cases due to loss of follow-up (38.6% of 228 patients), which is explained by the nature of the retrospective design and the referral structure at our clinic.

Our data could not detect prognostic effects of preoperative parameters such as foveal height of detachment, degrees of macular areas detached, shorter duration of macular detachment or visual loss until surgery, as is commonly established [1, 5, 6, 22, 23]. The subgroup without foveal detachment shows better baseline and final visual acuity, in accordance with these reports, but our subgroups were too small to make a statistical conclusion (S2 Table).

## Conclusion

In conclusion, we found a strong predictive power of one-month morphological and functional findings, namely the ELM integrity and ELM grade, for the final distance and reading visual acuity after 6 months. The technically simple qualitative assessment of the ELM state 1 month after retinal detachment surgery is able to predict the near and distance visual potential and may thus serve as an OCT biomarker in macula-involving RD, explaining 8% of the variability in distance and 29% in near VA after 6 months. While the prediction of final visual function from exams after 1 month remains difficult, a better state of the ELM after 1 month may indicate favorable final VA outcome, and ELM integrity may indicate favorable final RA.

## Supporting information

**S1 Table. Descriptive statistics of visual acuity by subgroups.** Descriptive statistics of visual acuity (VA; measured in LogMAR) by subgroups with or without central foveal detachment in pars plana vitrectomy for rhegmatogenous retinal detachment with macular involvement. (DOCX)

**S2 Table. Descriptive statistics of visual and reading acuity by subgroups.** Descriptive statistics of visual acuity (VA) and reading acuity (VA) by subgroups with or without ILM-peeling in pars plana vitrectomy for rhegmatogenous retinal detachment with macular involvement. (DOCX)

**S3 Table. Detailed description of multiple regression models with correlations and prognostic factors 1 month after retinal detachment surgery with regard to visual function after 6 months.**
(DOCX)

**S1 File. Anonymized full data on which the manuscript is based on.**
(XLSX)

## Acknowledgments

We thank the entire technical team of the Berner Augenklinik am Lindenhofspital, especially Sonja Steinhauer and Natalie Kaufmann, for the careful follow-up including the performing of OCT-imaging and reading acuity testing, which requires additional attention during the stressful everyday routine.

## Author Contributions

**Conceptualization:** Christof Hänsli, Christin Schild, Justus G. Garweg.

**Data curation:** Christof Hänsli, Suijana Lavan, Isabel B. Pfister, Christin Schild.

**Formal analysis:** Christof Hänsli, Isabel B. Pfister, Justus G. Garweg.

**Investigation:** Justus G. Garweg.

**Methodology:** Christof Hänsli, Isabel B. Pfister, Christin Schild, Justus G. Garweg.

**Project administration:** Christin Schild, Justus G. Garweg.

**Resources:** Justus G. Garweg.

**Supervision:** Christof Hänsli, Christin Schild, Justus G. Garweg.

**Validation:** Christof Hänsli, Isabel B. Pfister, Justus G. Garweg.

**Visualization:** Christof Hänsli, Isabel B. Pfister.

**Writing – original draft:** Christof Hänsli, Suijana Lavan, Isabel B. Pfister, Justus G. Garweg.

**Writing – review & editing:** Christof Hänsli, Suijana Lavan, Isabel B. Pfister, Christin Schild, Justus G. Garweg.

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
