## [Decision Letter · Decision Letter 0]

1 Mar 2022

PONE-D-21-40908Outer retinal features in OCT predict visual recovery after primary macula-involving retinal detachment repairPLOS ONE

Dear Dr. Hänsli,

Thank you for submitting your manuscript to PLOS ONE. After careful consideration, we feel that it has merit but does not fully meet PLOS ONE’s publication criteria as it currently stands. Therefore, we invite you to submit a revised version of the manuscript that addresses the points raised during the review process.

We look forward to receiving your revised manuscript.

Kind regards,

Akram Belghith

Academic Editor

PLOS ONE

Journal Requirements:

Reviewers' comments:

Reviewer's Responses to Questions

**Comments to the Author**

1. Is the manuscript technically sound, and do the data support the conclusions?

Reviewer #1: Yes

2. Has the statistical analysis been performed appropriately and rigorously? 

Reviewer #1: Yes

3. Have the authors made all data underlying the findings in their manuscript fully available?

Reviewer #1: Yes

4. Is the manuscript presented in an intelligible fashion and written in standard English?

Reviewer #1: Yes

5. Review Comments to the Author

Reviewer #1: The authors performed a retrospective analysis on their macular off retinal detachment cohort. The paper is well written and coherent. The topic is still of interest since macular off retinal detachment remains challenging and the prognostic factors for visual recovery are still not fully understood. On major point of criticism is the fact that most patients underwent ILM peeling during vitrectomy for retinal detachment. This was done due to personal preference of the surgeon but might not be considered as standard of care. Therefore, it remains unclear how the conclusion can be applied on patients that did not receive ILM peeling during surgery. The peeling itself may cause considerable changes to the inner retinal layers and these changes might cause longitudinal retinal layer thickness changes such as microcystic alterations. Regarding retinal layer changes after macula off retinal detachment repair, there is data already available, which might be discussed in more detail. (Invest Ophthalmol Vis Sci. 2014 Sep 4;55(10):6575-9.)

6. PLOS authors have the option to publish the peer review history of their article (what does this mean?). If published, this will include your full peer review and any attached files.

Reviewer #1: No

---

## [Author Response · Author response to Decision Letter 0]

6 Mar 2022

Dear Akram Belghith, dear Reviewer,

Thank you for your kind appreciation of the scientific and clinical value of our study in the still debated topic of retinal detachments with macular involvement. 

• As discussed in our manuscript and justifiably recalled by the reviewer, most patients underwent ILM peeling due to personal preference of the surgeon, which may not be universally accepted standard of care. The limited generalizability for patients without ILM peeling was added in the revised discussion. 

• Also highlighted by the reviewer ILM peeling may cause considerable changes to the retinal layers, with possibly longitudinal retinal layer thickness changes. The concern is addressed with an additional reference (Hisatomi et al, Retina 2018 Mar;38(3):471-479) in the revised discussion.

• The previously mentioned data of retinal layer changes after macula off retinal detachment repair has been formulated more explicitly and the suggested reference (Invest Ophthalmol Vis Sci. 2014 Sep 4;55(10):6575-9.) was added to the previously quoted literature. 

We believe that we have answered all the criticized points sufficiently and implemented them in the revised manuscript, highlighted in the track-changed version. Also, the additional style guides have been implemented in the revised manuscript. 

Thank you for considering the revised manuscript of our study for publication in PLOS one.

Sincerely yours,

Christof Hänsli

---

## [Editor Report · Decision Letter 1]

21 Apr 2022

Outer retinal features in OCT predict visual recovery after primary macula-involving retinal detachment repair

PONE-D-21-40908R1

Dear Dr. Hänsli,

We’re pleased to inform you that your manuscript has been judged scientifically suitable for publication and will be formally accepted for publication once it meets all outstanding technical requirements.

Kind regards,

Akram Belghith

Academic Editor

PLOS ONE

---

## [Editor Report · Acceptance letter]

26 Apr 2022

PONE-D-21-40908R1 

Outer retinal features in OCT predict visual recovery after primary macula-involving retinal detachment repair 

Dear Dr. Hänsli:

I'm pleased to inform you that your manuscript has been deemed suitable for publication in PLOS ONE. Congratulations! Your manuscript is now with our production department. 

Kind regards, 

on behalf of

Dr. Akram Belghith 

Academic Editor

PLOS ONE